# The Synergistic Effect of Zuogui Pill and Eldecalcitol on Improving Bone Mass and Osteogenesis in Type 2 Diabetic Osteoporosis

**DOI:** 10.3390/medicina59081414

**Published:** 2023-08-03

**Authors:** Tuo Shi, Ting Liu, Yuying Kou, Xing Rong, Lingxiao Meng, Yajun Cui, Ruihan Gao, Sumin Hu, Minqi Li

**Affiliations:** 1School of Traditional Chinese Medicine, Beijing University of Chinese Medicine, Beijing 100029, China; 15652386608@163.com; 2Department of Bone Metabolism, School and Hospital of Stomatology, Cheeloo College of Medicine, Shandong University & Shandong Key Laboratory of Oral Tissue Regeneration & Shandong Engineering Laboratory for Dental Materials and Oral Tissue Regeneration & Shandong Provincial Clinical Research Center for Oral Diseases, Jinan 250012, China; 18054504397@163.com (T.L.); kouyuying2023@163.com (Y.K.); rongxing797@163.com (X.R.); mlx15376409774@163.com (L.M.); cuiyajunsdu@163.com (Y.C.); diting12138@163.com (R.G.); 3Center of Osteoporosis and Bone Mineral Research, Shandong University, Jinan 251600, China

**Keywords:** Zuogui pill, eldecalcitol, type 2 diabetic osteoporosis, osteoblast, PI3K–AKT signaling pathway

## Abstract

*Background and Objectives:* The incidence of diabetic osteoporosis, an important complication of diabetes mellitus, is increasing gradually. This study investigated the combined effect of the Zuogui pill (ZGP) and eldecalcitol (ED-71), a novel vitamin D analog, on type 2 diabetic osteoporosis (T2DOP) and explored their action mechanism. *Materials and Methods:* Blood glucose levels were routinely monitored in db/db mice while inducing T2DOP. We used hematoxylin and eosin staining, Masson staining, micro-computed tomography, and serum biochemical analysis to evaluate changes in the bone mass and blood calcium and phosphate levels of mice. Immunohistochemical staining was performed to assess the osteoblast and osteoclast statuses. The MC3T3-E1 cell line was cultured in vitro under a high glucose concentration and induced to undergo osteogenic differentiation. Quantitative real-time polymerase chain reaction, Western blot, immunofluorescence, ALP, and alizarin red staining were carried out to detect osteogenic differentiation and PI3K–AKT signaling pathway activity. *Results:* ZGP and ED-71 led to a dramatic decrease in blood glucose levels and an increase in bone mass in the db/db mice. The effect was strongest when both were used together. ZGP combined with ED-71 promoted osteoblast activity and inhibited osteoclast activity in the trabecular bone region. The in vitro results revealed that ZGP and ED-71 synergistically promoted osteogenic differentiation and activated the PI3K–AKT signaling pathway. The PI3K inhibitor LY294002 or AKT inhibitor ARQ092 altered the synergistic action of both on osteogenic differentiation. *Conclusions:* The combined use of ZGP and ED-71 reduced blood glucose levels in diabetic mice and promoted osteogenic differentiation through the PI3K–AKT signaling pathway, resulting in improved bone mass. Our study suggests that the abovementioned combination constitutes an effective treatment for T2DOP.

## 1. Introduction

Diabetes mellitus (DM), a severe chronic metabolic illness, has spread rapidly in recent years. More than 425 million people were reported to have DM in the 2017 worldwide statistics of the International Diabetes Federation Atlas, a number that could increase to 156% by 2045 [1,2]. Type 2 DM (T2DM) is common among adults and older adults, accounting for approximately 90% of patients with DM [3]. Type 2 diabetic osteoporosis (T2DOP), a T2DM complication, is a systemic bone illness marked by low bone mass and deterioration of the bone microstructure, resulting in brittle bones prone to fractures [4]. Evidence supporting an association between T2DM and the susceptibility to fractures is growing. The incidence of secondary osteoporosis (OP) and the risk of fractures in people with DM is 4–5 times higher than in the general population [5]. The diabetic osteoporosis (DOP) incidence in patients with T2DM is as high as 20–60% [6]. Therefore, developing effective therapeutic agents for T2DOP could have considerable economic and social significance.

Bone homeostasis depends on osteoclast-mediated bone resorption and osteoblast-mediated bone formation. Osteoblasts are involved in bone matrix synthesis and mineral deposition. The main characteristics of T2DOP are reduced bone turnover and impaired osteoblast activity [7,8,9], mainly due to impaired osteoblast proliferation, differentiation, and function due to persistent hyperglycemia, the formation of advanced glycation end products (AGEs), oxidation products, insulin deficiency, obesity, and microangiopathies in T2DM [10,11,12]. A high-glycemic environment is often used to model T2DOP in vitro and help elucidate its pathogenesis [13,14].

The Zuogui pill (ZGP) is a well-known traditional Chinese medicine (TCM) prescription derived from the medical book *The Complete Works of [Zhang] Jing-yue of the Ming Dynasty*. It comprises eight Chinese herbs, namely *Rehmannia glutinosa* Libosch. (prepared root); *Lycium barbarum* L. (dried fruit); *Dioscorea opposita* Thunb. (dried rhizome); *Cyathula officinalis* Kuan (dried root); *Cornus officinalis* Sieb. et Zucc. (dried fruit); *Cervus elaphus* Linnaeus (antler); *Cuscuta chinensis* Lam. (dried seed); and *Chinemys reevesii* (Gray) (shell) (8:4:4:3:4:4:4:4) [15]. According to TCM, the kidney, which is one of the five zang organs, has one of the functions of governing bones. The kidney stores the yin essence, which produces and generates marrow, while the marrow fulfills and nourishes bones. Therefore, when the kidney’s yin essence is deficient, and there is not enough nourishment, bone loss, or osteoporosis, occurs [16]. Thereby, nourishing kidney yin is considered a therapeutic method for osteoporosis [17]. Due to its function of nourishing kidney yin and replenishing this essence, ZGP is recommended by the Chinese Association of Integrative Medicine (Chinese Association of Integrative Medicine, 2019) and the Chinese Association of Chinese Medicine (Chinese Association of Chinese Medicine, 2020) as a treatment drug in the recently updated OP clinical guidelines [18]. ZGP promotes osteoblast differentiation [19,20,21] and significantly increases bone density in senile OP, postmenopausal OP [21], and glucocorticoid-induced OP (GIOP) [16,22]. Furthermore, ZGP treatment positively affected DM by decreasing the glucose level in a rat model [23], avoiding the adverse effects of high sugar on embryonic development [24], and preventing impaired glucose tolerance in the offspring induced by a high sugar and fat diet. Conversely, ZGP improves kidney injury in diabetes [25]. However, its effect on DOP has yet to be reported. Due to its unique advantages as a TCM, the ZGP effect improves when combined with Western medicine. For example, a study found that ZGP combined with anti-OP drugs, such as alendronate, led to better bone mineral density recovery than alendronate alone [18], suggesting that such combinations could effectively treat DOP.

Vitamin D (VD) is crucial for bone tissue development and remodeling [26]. It maintains the body’s calcium and phosphate balance and regulates intestinal and renal reabsorption. VD also affects mineralization, bone conversion rate, and fracture occurrence by acting on various bone tissue cells and factors, thereby contributing to OP prevention and treatment [27]. The VD level in patients with T2DM affects the blood sugar level and bone metabolism. Insulin resistance and bone turnover were reported to be inversely linked to the 25(OH)D concentration. A drop in the 25(OH)D concentration and an increase in insulin resistance in patients with T2DM triggered bone turnover and increased the risk of OP [28].

The liver and kidneys metabolize the activated form of VD (AVD). Eldecalcitol (ED-71) is an emerging AVD analog with a longer half-life and a lower hypercalcemia risk than previous AVDs [29]. More importantly, ED-71 has exhibited superior therapeutic efficacy against various OP types. In an earlier study, ED-71 stimulated pre-osteoblast differentiation and downregulated osteoclast number, thus reducing bone loss in OVX rats [30]. In GIOP mice, ED-71 prevented glucocorticoid-induced bone loss by promoting osteogenic differentiation [31]. In a mouse DOP model, ED-71 promoted M2 macrophage polarization and induced the osteogenic differentiation of BMSCs [32]. ED-71 has also achieved good results when combined with other drugs. Cotreatment with exendin-4 had a remarkable glucose-reducing effect in DOP mice and enhanced osteogenesis [32]. ED-71 sequential treatment after PTH inhibited PTH withdrawal-associated OP in estrogen-deficient rats [33]. There is no relevant study on the combined effect of ZGP and ED-71 on OP.

This study investigated whether the ZPG and ED-71 combination could improve T2DOP and explored the associated specific mechanisms. We also verified that ZPG and ED-71 cotreatment promoted osteogenic differentiation in a PI3K–AKT-dependent manner. The study results could help open new venues for T2DOP treatment.

## 2. Materials and Methods

### 2.1. Reagents

ED-71 (Chugai Pharmaceutical Co., Ltd., Tokyo, Japan) was dissolved in 100% ethanol for storage and diluted to the desired concentration in medium-chain triglycerides (MCTs) before usage. MCTs with the same absolute ethanol concentration were used as controls. ZGPs were purchased from Beijing Tongrentang Co., Ltd., Beijing, China. The granules were used to prepare a ZGP suspension at the desired concentration, which was kept at 4 °C.

The anti-Runx2 (20700-1-AP) and anti-GAPDH (10494-1-AP) antibodies were purchased from Proteintech (Chicago, IL, USA). The anti-osteocalcin (OCN) antibody (bs-4917R) was purchased from Bioss (Beijing, China). The anti-alkaline phosphatase (ALP) (ab65834), anti-P-PI3K (ab278545), and anti-P-AKT (ab192623) antibodies as well as goat anti-rabbit (ab6721), goat anti-mouse (SA00001-1), and goat anti-rabbit (SA00013-2) IgGs were purchased from Abcam (Cambridge, UK). The ARQ092 (HY-19719) and LY294002 (HY-10108) inhibitors were purchased from MedChemExpress (Monmouth Junction, NJ, USA).

### 2.2. Animals and Drug Administration

Male C57BLKS/J Iar-+Leprdb/+Leprdb mice (db/db; 33–45 g; 8 w; *n* = 5), as a DOP model, were acquired from the Model Animal of Nanjing University (Jiangsu, China). The mice were maintained under normal laboratory conditions at 20 °C and subjected to a normal 12 h/12 h light and dark routine. We randomly divided the animals into four groups: control (PBS given orally for 6 w); ZGP (3.8 g/kg/d ZGP given orally for 6 w); ED-71 (0.25 μg/kg ED-71 given orally 3 times/w for 6 w [32]); ZGP + ED-71 (3.8 g/kg/d ZGP and 0.25 μg/kg ED-71 3 times/w given orally for 6 w).

### 2.3. Blood Glucose Assay and Serum Biochemical Analysis

The mice were deprived of food and water for 8 h/w before being pricked in the tail vein to measure the blood glucose level with a blood glucose meter (China Yuwei Medical Equipment Co., Ltd., Shenzhen, China). The mice were anesthetized, and blood samples were collected from their posterior orbit before they were sacrificed. Serum samples were separated via centrifugation at 3000 rpm for 10 min and stored at −80 °C. A fully automatic biochemical analyzer (BS-240VET, Mindray, Shenzhen, China) was used to measure the serum levels of calcium and phosphate using dedicated kits (Century World Biotechnology Co., Ltd., Shenzhen, China), according to the manufacturer’s instructions.

### 2.4. Animal Euthanasia and Sample Preparation

Mice were euthanized via anesthetic overdose. Their tibia bones were placed in 4% paraformaldehyde phosphate-buffered saline with 10% EDTA-2Na for one month at 4 °C.

### 2.5. Micro-Computed Tomography (CT) Scan

Samples were dissected aseptically and scanned, and the data were reconstructed using a micro-CT system (Scanco Medical, Wangen-Brüttisellen, Switzerland) to generate three-dimensional images.

### 2.6. Hematoxylin and Eosin (HE) Staining

The sections were dewaxed through immersion in des-gradient alcohol, stained with hematoxylin for 15 min, washed with distilled water, stained with eosin for 7 min, and washed again with distilled water. Mounts were observed on an optical microscope (Olympus BX-53, Tokyo, Japan), and images were acquired.

### 2.7. Masson Staining

After dewaxing and hydration, the sections were stained with hematoxylin for 10 min, placed in a Ponceau S acid solution, allowed to undergo differentiation in a phosphomolybdic solution, and then transferred to an aniline blue solution. The mounts were observed on an optical microscope (Olympus BX-53, Tokyo, Japan).

### 2.8. Immunohistochemical and Tartrate-Resistant Acid Phosphatase (TRAP) Staining

Tissue sections were immersed in alcohol, followed by treatment with 1% bovine serum albumin (BSA) in PBS for 20 min, to block nonspecific staining. The sections were incubated with primary antibodies overnight at 4 °C, washed with PBS, incubated with secondary antibodies at room temperature for 1 h, washed in 3, 3′-diaminobenzidine tetrahydrochloride, and counterstained with methyl green. The sections were washed in double-distilled water and immersed in the TRAP staining solution and then restained in methyl green. The sealed sections were observed under optical microscopy (Olympus Corp), and digital images were captured.

### 2.9. Cell Culture, Acquisition of Medicated Serum, and Osteogenic Differentiation Induction

Mouse pre-osteoblast MC3T3-E1 cells from Shanghai Cell Center (Shanghai, China) were incubated with an α-minimal medium with 10% serum, 100 units/mL penicillin, 100 μg/mL streptomycin, and a high glucose environment of 35 mmol/L [18] at 37 °C and 5% CO_2_. Medicated serum was acquired from eight-week-old male Sprague Dawley rats [34]. We randomly divided them into four groups: control (PBS given orally for 7 d); ZGP (ZGP 1.85 g/kg/d, given orally for 7 d); ED-71 (ED-71 0.125 μg/kg, given orally 3 times/w); and ZGP + ED-71 (ZGP 3.8 g/kg/d and ED-71 0.25 μg/kg 3 times/w, both given orally). The rats’ blood was withdrawn after 7 d [34], filtered, aliquoted, and stored at −80 °C.

The cells were cultured for 7 or 21 days in an osteogenesis-inducing medium comprising 50 mg/L ascorbic acid, 100 mM β-glycerophosphate, 10 nM dexamethasone, and drug-containing serum in a high-glucose environment of 35 mmol/L. We changed this medium every 3 d.

### 2.10. Alkaline Phosphatase and Alizarin Red (AR) Staining

After inducing osteogenesis for 7 d, the cells were fixed in 4% paraformaldehyde for 20 min, stained with an ALP solution for 30 min, and washed in PBS. ALP-positive cells and calcified nodules were observed and analyzed using Image Pro Plus 6.0 (IPP 6.0) software. Other batches of cells were fixed in 4% paraformaldehyde, stained with 1% AR staining solution (Solarbio, Beijing, China), and washed in PBS. The AR-stained nodules were separated using cetylpyridinium chloride dissolved in PBS and detected using a spectrophotometer at 450 nm.

### 2.11. Immunofluorescence Staining

Cells were fixed in 4% paraformaldehyde, permeated with 0.5% Triton X-100, and blocked with 5% BSA in PBS. Subsequently, they were incubated with a primary antibody at 4 °C overnight and then with a secondary antibody at room temperature for 1 h. Finally, the cells were counterstained with DAPI for 5 min, and the relative fluorescence intensity was measured using ImageJ software.

### 2.12. Quantitative Real-Time Polymerase Chain Reaction (RT-qPCR) Analysis

The total RNA was extracted from cell samples using the RNAex Pro Reagent (Accurate Biology, Changsha, China) and reverse-transcribed into cDNA using an Evo M-MLV Reverse Transcription Kit (Accurate Biology, Changsha, China). RT-qPCR was performed in triplicate with the SYBR Green PCR kit (Accurate Biology, Changsha, China) using the Roche Light Cycler 96 Real-time PCR system (Roche, Welwyn Garden City, UK). The relative expression levels of *RUNX2*, *ALP*, *OCN*, and *GAPDH* were calculated using the 2^−ΔΔCt^ method. The levels of the first three factors were normalized to that of *GAPDH*. The primer sequences are listed in Table 1.

### 2.13. Western Blotting

Total cell proteins were extracted in a mixture containing phosphatase inhibitor, protease inhibitor, and RIPA lysis buffer (Cwbio, Beijing, China) at a ratio of 98:1:1. The protein concentration was assessed using a BCA Assay Kit (Beyotime, Shanghai, China). A standard quantity of proteins was electrophoresed on SDS-PAGE and transferred to a PVDF membrane. The membrane was incubated overnight with a primary antibody at 4 °C and then with a secondary antibody for 1 h. The bands were measured using an enhanced chemiluminescence reagent (Proteintech, Chicago, IL, USA) and an ECL system (General Electric Company, Boston, MA, USA). Each experiment was repeated in triplicate.

### 2.14. Statistical Analysis

Data were approximately normally distributed. All quantitative data are expressed as means ± standard deviations (SDs), and all experiments were independently repeated at least 3 times. One-way ANOVA was used for multiple groups’ comparison, and the mean value of each group was compared using the least significant difference (LSD) test. Statistical analysis was performed using GraphPad Prism 6.0 software (GraphPad Software, San Diego, United States), and *p* < 0.05 was considered statistically significant.

## 3. Results

### 3.1. ZGP and ED-71 Synergistically Reduced Blood Glucose Levels in Diabetic Mice

The blood glucose levels in the db/db mice were markedly higher than in the wild-type (WT; Figure 1A), while ZGP or ED-71 treatment reduced the blood glucose level. ED-71 exhibited a stronger effect than ZGP in the early administration stage, whereas ZGP exhibited a more significant effect from the third week onward. More importantly, treatment with both showed a stronger sugar-lowering effect than each alone (Figure 1B). The liver tissue’s HE staining revealed no significant difference between the treatment and control groups, suggesting that ZGP and ED-71 caused no damage to the liver. Meanwhile, microvesicular liver steatosis was reduced after drug treatment (Figure 1C).

### 3.2. ZGP and ED-71 Synergistically Increased Bone Mass, Improved Osteoblasts, and Inhibited Osteoclasts in Diabetic Mice

Micro-CT revealed that the db/db mice had very little cancellous bone. The ZGP treatment slightly increased the cancellous bone mass, whereas the ED-71 treatment increased it significantly, and the ZGP + ED-71 combination exhibited the strongest effect (Figure 2A). HE staining revealed that the number, volume, and width of the trabecular bone increased, and the separation was reduced after ZGP or ED-71 treatment. These observations were more significant after administering the ZGP + ED-71 combination than any alone (Figure 2B,C). Masson staining also demonstrated that the ZGP + ED-71 combination increased bone mass, as indicated by increased blue-stained new bone regeneration regions (Figure 2B). Serum calcium and phosphor levels were in the following increasing order: control, ZGP, ED-71, and ED-71 + ZGP (Figure 2D).

Osteoblasts and osteoclasts in the metaphyseal region of the mice were observed histologically. ZGP or ED-71 improved the positive expression of ALP and COL-1, increased the number of RUNX2-positive osteoblasts, reduced the number of TRAP-positive osteoclasts, and inhibited the positive expression of MMP-9. This suggested that ZGP and ED-71 could stimulate the osteoblasts and inhibit the osteoclasts (Figure 3A,B). This effect was further enhanced when both were administered together (Figure 3A,B).

### 3.3. The ZGP and ED-71 Combination Promoted Osteogenic Differentiation of the MC3T3-E1 Cells

After the osteogenic induction of the MC3T3-E1 cells in a high-glucose environment, Western blot revealed that ZGP and ED-71 increased ALP, RUNX2, and OCN protein levels, with the largest increase achieved when both were administered together (Figure 4A). RT-qPCR analysis demonstrated a similar trend, with the highest increase in *ALP*, *RUNX2*, and *OCN* mRNA expression in the cotreatment group (Figure 4A). ALP staining revealed that ALP expression was enhanced in the ZGP and ED-71 groups and was the strongest in the combination group (Figure 4B). AR staining demonstrated that treatment with ZGP or ED-71 increased the calcified nodule number, which was the highest in the combination group (Figure 4B). These results demonstrated that the ZGP and ED-71 combination promoted osteogenic differentiation and mineralization in the MC3T3-E1 cells.

### 3.4. ZGP Synergized with ED-71 to Promote the Osteogenic Differentiation of MC3T3-E1 Cells through the PI3K–AKT Pathway

The specific mechanism through which ZGP synergizes with ED-71 to promote osteogenic differentiation was explored. The P-PI3K and P-AKT expression levels gradually increased in the control, ZGP, ED-71, and ZGP + ED-71 groups (Figure 5A). The PI3K inhibitor LY294002 or the AKT inhibitor ARQ092 validated the signaling pathway activity. The fluorescence intensity of P-PI3K and P-AKT in the MC3T3-E1 cells, which increased when treated with the ZGP + ED-71 combination, was inhibited by LY294002 (Figure 5B). When ARQ092 was added, the expression of P-AKT was inhibited, while that of P-PI3K was not (Figure 5B). RT-qPCR and Western blotting revealed that LY294002 addition decreased *ALP*, *OCN*, and *RUNX2* mRNA and protein expression and reversed the promoting effect of ZGP and ED-71 on osteoblast differentiation (Figure 6A,B). Similarly, the osteoblast differentiation-promoting effect of ZGP and ED-71 was reversed when AKT was inhibited by ARQ092 (Figure 6C,D). ALP staining revealed that LY294002 and ARQ092 inhibited the positive expression of ALP induced by the ZGP and ED-71 combination (Figure 6E).

## 4. Discussion

We evaluated the therapeutic effects of ZGP and ED-71 on T2DOP, their ability to improve osteogenic differentiation in a high-glycemic environment, and their potential mechanism of action. Our results demonstrated that ZGP and ED-71 synergistically improved T2DOP by increasing the bone mass and osteogenesis rate. These positive effects were, at least in part, achieved by promoting osteoblast differentiation. PI3K–AKT signaling pathway activation is a potential mechanism through which ZGP and ED-71 promote osteoblast differentiation.

T2DM, a widespread metabolic disease, is typically manifested in persistent hyperglycemia. Accumulating evidence supports the close association between high-glucose levels and alterations in bone metabolism [35,36,37]. The reduction in blood glucose levels has become a major challenge that needs to be addressed. In this study, the blood glucose level of db/db mice was reduced through the ZGP or ED-71 treatment, an effect that was more obvious when both drugs were administered together. Although the blood sugar-lowering effect of ZGP was slow, it exhibited a more significant, stable, and lasting effect from the third week onward. Studies have reported that ZGP can effectively lower blood sugar levels [24,25,38,39]. ZGP consists of eight Chinese herbs. The *Rehmannia glutinosa* Libosch. and *Dioscorea opposita* Thunb. in the ZGP can, respectively, inhibit α-amylase and α-glucosidase, two crucial enzymes involved in glucose production, leading to a significant reduction in the post-prandial glucose level [40,41]. *Lycium barbarum* L. is rich in betaine, which is critical in lowering blood sugar levels [42]. Quercetin-3-o-galactoside, an active flavonoid glycoside, primarily obtained from *Cuscuta chinensis* Lam., has been shown to dramatically reduce the fasting blood glucose levels in db/db mice [43]. The observed long-term blood glucose-reducing function of ZGP could be due to the many components of ZGP that exert glucose-reducing effects through multiple targets. VD supplements effectively reduced the T2DM risk [44,45]. We found in this study that ED-71 effectively reduced the blood glucose levels in db/db mice, consistent with previous reports [32]. Notably, ED-71 exhibited a rapid hypoglycemic effect in the early stage of administration, possibly because it is an AVD analog and can function directly without the traditional VD activation process [31]. Interestingly, we found that the blood glucose level reduction was higher when the ZGP and ED-71 combination was administered than when either drug was given alone. Therefore, we speculated that the two-drug combination triggered a full play of their respective functions to achieve the best hypoglycemic effect. Considering the fact that previous studies have shown that a high glucose environment can inhibit the osteogenic differentiation and activity of osteoblasts as well as increase the osteoclast differentiation and osteoclast activity [46], we believe that the cotreatment may improve bone mass indirectly by downregulating the blood glucose levels.

Patients with DOP exhibit significant bone mass reduction, bone turnover reduction, and impaired osteoblast activity [8,47]. A randomized controlled trial revealed that ZGP improved bone density in 55–75 years old patients with OP [48]. Furthermore, ZGP improves GIOP and dexamethasone-induced OP by improving the bone mineral extent, enhancing bone biomechanical strength, increasing osteogenesis, and hindering bone resorption [15,16,22,49]. It has been reported that the *Rehmannia glutinosa* Libosch. in the ZGP can regulate alkaline phosphatase activity and osteocalcin levels in diabetic rats, increase bone mineral density, and improve bone microstructure. The extract of the *Dioscorea opposita* Thunb. and *Lycium barbarum* L. in the ZGP increased bone mass and bone strength in models and promoted the proliferation, differentiation, and ossification of osteoblasts. Our results indicated that ZGP promoted osteogenesis and increased bone mass, consistent with previous reports indicating that ZGP improves OP. Furthermore, the ZGP and ED-71 combination exhibited the strongest effect in improving bone mass, stronger than ZGP or ED-71 alone. As an effective therapeutic agent for DOP [32,50], ED-71 enhanced the therapeutic effect of ZGP to increase bone mass and promote new bone formation in patients with DOP. Hypercalcemia is the most widespread complication of ED-71 [51]. Studies have also shown that ZGP treatment significantly increased serum calcium levels in rats [49]. Meanwhile, the same trend of increase in serum calcium levels was observed during ZGP and ED-71 treatment in our study. We found that the increase in the blood calcium level was more pronounced in the ZGP + ED-71 group, so serum calcium concentration should be monitored during clinical treatment.

The osteoblastic differentiation ability in patients with DOP is weakened in a high-glucose microenvironment [12], while the differentiation ability of osteoclasts is upregulated [52]. Current strategies used to treat DOP include osteogenic differentiation promotion of osteoblasts to increase bone mass in these patients [53]. Targeted osteogenic differentiation is a powerful treatment approach for T2DM. In this study, ALP, RUNX2, and COL-1 levels in the tibial metaphysis increased after administering ZGP or ED-71 to diabetic mice. The ZGP and ED-71 combination further increased the expression levels of these factors. In vitro experiments showed similar results, with the ZGP and ED-71 combination decreasing the blocking function exerted by the high glucose concentration on osteoblast differentiation. Increased ALP levels suggested enhanced osteoblast differentiation and increased osteogenesis [54]. The change in RUNX2 level indicated changes in osteoblast differentiation and mineralization [55]. OCN, a noncollagenous protein, is vital in bone matrix mineralization [56]. COL-1 is a marker of osteoblast differentiation and maturation [57]. ED-71 exerted a protective effect on osteoblasts when treating T2DM. Combined with exendin-4, it promoted M2 macrophage polarization and induced osteogenic differentiation of BMSCs in diabetic mice [32], consistent with our findings. ED-71 also upregulated osteoblast differentiation by improving osteoglycin secretion from myoblasts [58]. ZGP mitigated OP by enhancing osteogenic differentiation of the BMSCs [21]. This medication significantly promoted mineralized nodule formation and increased ALP and COL-1 expression to modulate the osteogenic differentiation of BMSCs in OVX rats [21]. We revealed for the first time that the ZGP and ED-71 combination could regulate osteogenic differentiation in a hyperglycemic state. Our results also showed that the ZGP and ED-71 combination significantly decreased the number of TRAP-positive osteoclasts and the expression of MMP-9, an enzyme secreted by osteoclasts, suggesting that the combined treatment could inhibit the osteoclasts and bone resorption in DOP. Our study illustrated that the combined treatment increased bone mass and attenuated the osteoporotic imbalance between bone formation and resorption by promoting osteoblasts and inhibiting osteoclasts. This combined therapy might be an effective treatment alternative for patients with DOP. In addition, while this study focused on osteoblasts, the effects of ZGP and ED-71 on osteoclasts will be explored in future research.

When exploring the mechanism, we focused on the PI3K–AKT pathway. PI3K is a crucial lipoprotein kinase controlling the proliferation, survival, and movement of cells after ligand activation [59], while AKT is a direct downstream target of PI3K [60]. The PI3K–AKT signaling pathway is crucial to osteoblast survival and differentiation [61]. Dexamethasone inhibits osteoblast differentiation in a PI3K–AKT-dependent manner [62]. PI3K–AKT is also related to OP, and it has been predicted to possess a crucial function in DOP pathogenesis [63]. According to our results, the ZGP and ED-71 promoted P-PI3K and P-AKT expression, suggesting that both could activate the PI3K–AKT signaling pathway. Importantly, they exhibited a better function when administered together. AVD could promote the PI3K–AKT signaling pathway in osteoblasts in a high-glucose state to affect autophagy [64]. ED-71 upregulated the PI3K–AKT signaling pathway in macrophages and promoted M2 polarization [32]. Currently, reports on the relationship between ZGP and the PI3K–AKT signaling pathway are lacking. Our results demonstrated the synergistic function of ZGP and ED-71 in activating the PI3K–AKT signaling pathway and osteoblast differentiation in a high-glucose state. This function was inhibited after blocking the PI3K–AKT signal with specific inhibitors. P-AKT expression was also reduced after using the PI3K inhibitor, demonstrating that AKT acts as a downstream signal for PI3K, consistent with previous research [53]. The PI3K–AKT signaling pathway has been suggested as an effective OP therapeutic target [60,65]. Our results indicated that ZGP and ED-71 could synergistically affect T2DOP by upregulating the PI3K–AKT signaling pathway. The PI3K–AKT signaling pathway was reported to be related to osteoclast differentiation [65]. The ZGP and ED-71 combination may affect the osteoclasts through this signaling pathway; however, further studies are still needed.

## 5. Conclusions

In our study, the ZGP and ED-71 combination reduced the blood glucose level, promoted osteoblast differentiation, inhibited osteoclasts, and increased bone mass in diabetic mice by regulating the PI3K–AKT signaling pathway. The superior therapeutic function of the ZGP and ED-71 combination lays a foundation for exploring a new clinical therapy for DOP. This combination treatment is expected to be effective in promoting bone health, decreasing levels of blood glucose, and reducing fracture risk in patients with DOP.

## Figures and Tables

**Figure 1 medicina-59-01414-f001:**
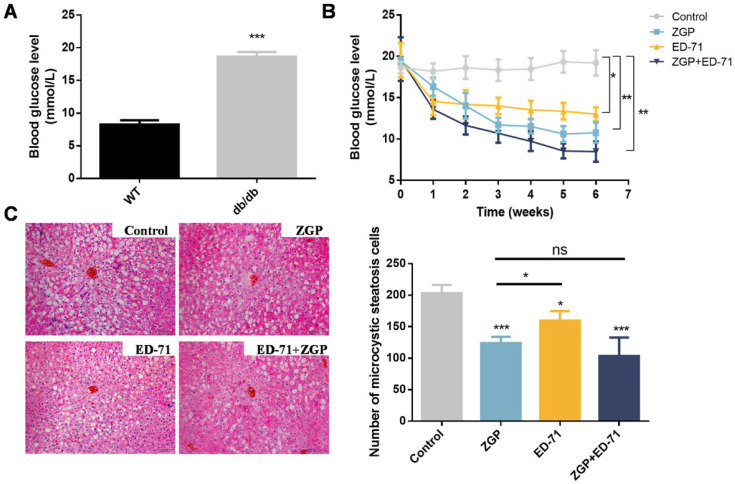
ZGP synergizes with ED-71 to reduce the blood glucose level in db/db mice: (**A**) comparison of blood glucose levels between WT and db/db mice (*n* = 6); (**B**) changes in blood glucose levels during the treatment (*n* = 6); (**C**) HE staining. Bar, 100 μm. The values are expressed as means ± standard deviations. ns: no significance, * *p* < 0.05, ** *p* < 0.01, and *** *p* < 0.001.

**Figure 2 medicina-59-01414-f002:**
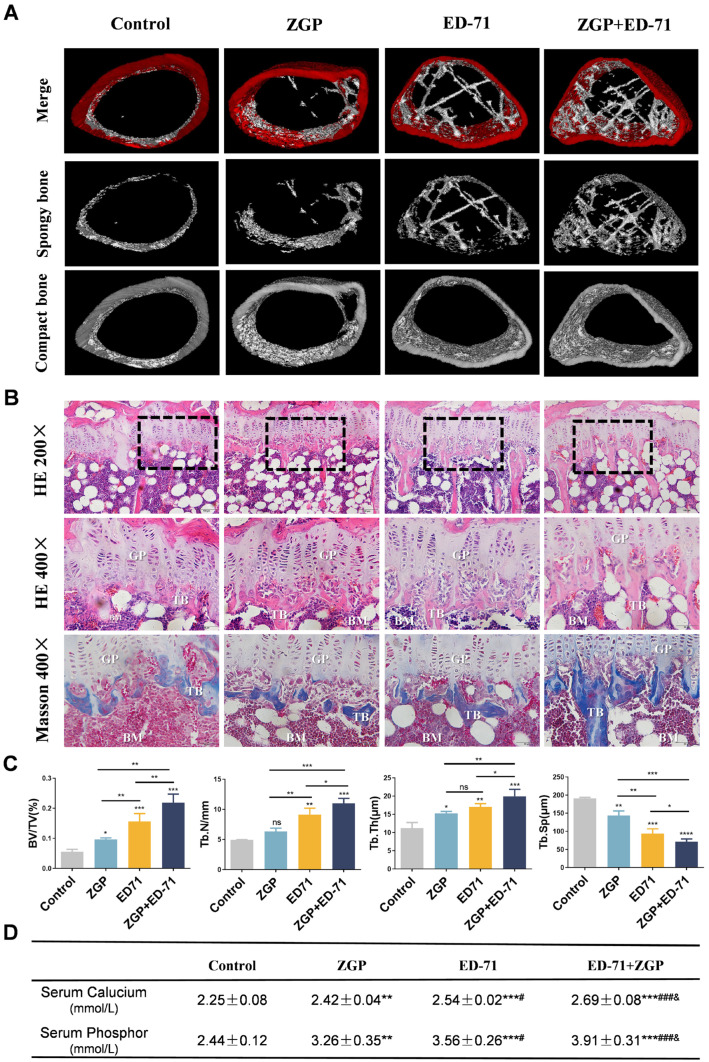
ZGP synergizes with ED-71 to increase bone mass in db/db mice: (**A**) representative 3D micro-CT images of the femur in the CON, ZGP, ED-7I, and ZGP + ED-71 groups after 6 w of treatment; (**B**) HE and Masson staining of the mice tibia. Bar, 50 or 100 μm; (**C**) bone/tissue volume, and trabecular bone number, thickness, and space were analyzed via HE staining (*n* = 3); (**D**) serum calcium and phosphor levels (*n* = 3); ** *p* < 0.01; *** *p* < 0.001 VS. Control Group; ^#^
*p* < 0.05; ^###^
*p* < 0.001 VS. ZGP Group; ^&^
*p* < 0.05 VS. ED-71 Group. The values are expressed as means ± SDs. ns: no significance, * *p* < 0.05, ** *p* < 0.01, and *** *p* < 0.001, **** *p* < 0.0001. BV/TV, bone/tissue volume; Tb.N, trabeculae bone number; Tb.Th, trabeculae bone thickness; Tb.Sp, trabeculae bone space; GP: growth plate; TB: trabeculae bone; BM: bone marrow.

**Figure 3 medicina-59-01414-f003:**
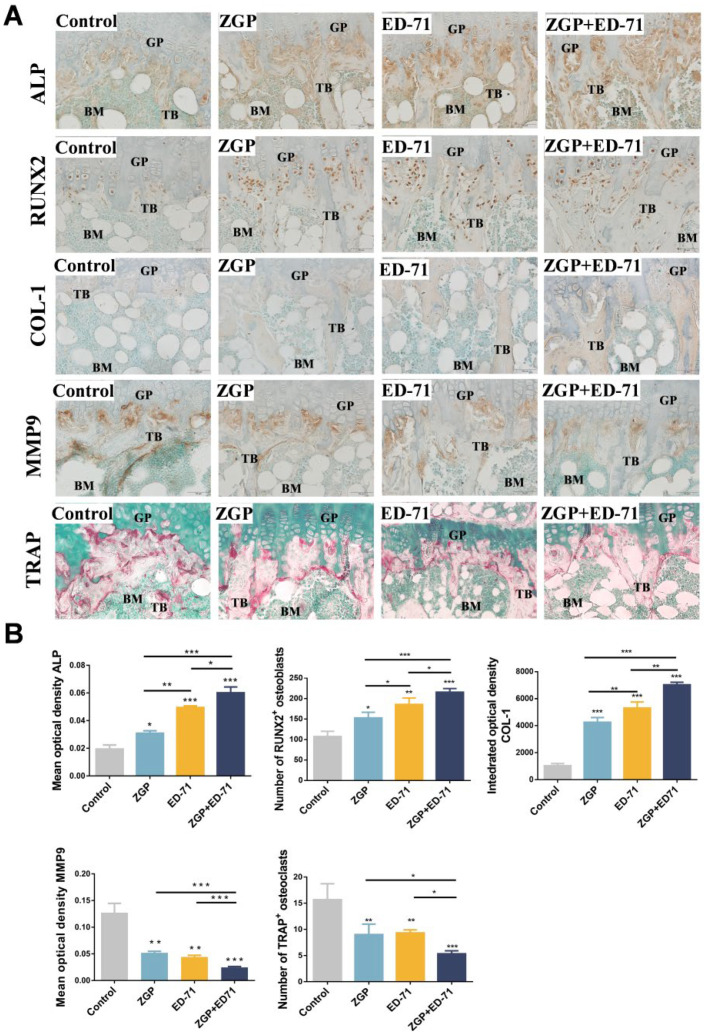
ZGP synergizes with ED-71 to stimulate the osteoblasts in db/db mice: (**A**) TRAP staining and immunohistochemical staining of ALP, RUNX2, COL-1, and MMP-9. Bar, 50 μm; (**B**) statistical analysis of the mean optical density of ALP and MMP-9, the integrated optical density of COL1, and the number of RUNX2-positive osteoblasts and TRAP-positive osteoclasts. The data are presented as means ± SDs (*n* = 3). * *p* < 0.05, ** *p* < 0.01, and *** *p* < 0.001. GP: growth plate; TB: trabeculae bone; BM: bone marrow.

**Figure 4 medicina-59-01414-f004:**
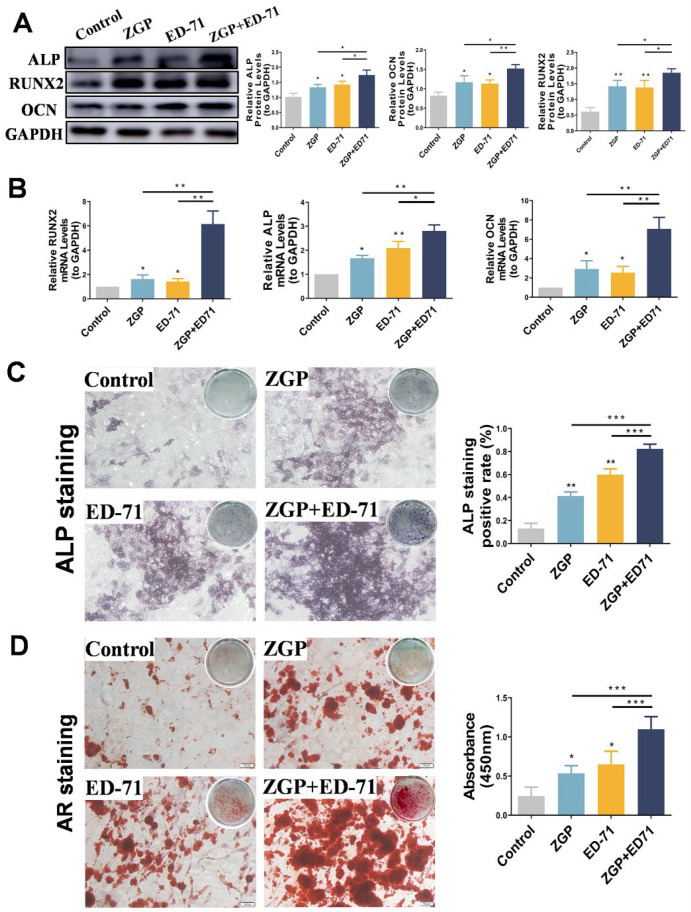
ZGP synergizes with ED-71 in affecting the differentiation and mineralization of HG-treated MC3T3-E1 cells: (**A**) the protein expression of ALP, RUNX2, OCN, and GAPDH was detected using Western blotting after 14 days of culture; (**B**) the mRNA expression of *ALP*, *RUNX2*, *OCN*, and *GAPDH* was assessed via RT-qPCR after seven days of culture; (**C**) ALP staining of MC3T3-E1 cells in the CON, ZGP, ED-71, and ZGP + ED-71 groups after seven days of culture, and the quantitative statistical analysis of ALP staining; (**D**) AR staining of MC3T3-E1 cells in the CON, ZGP, ED-71, and ZGP + ED-71 groups after 21 days of culture, and the quantitative statistical analysis of AR staining. Bar, 100 μm. The data are presented as means ± SDs (*n* = 3). * *p* < 0.05, ** *p* < 0.01, and *** *p* < 0.001.

**Figure 5 medicina-59-01414-f005:**
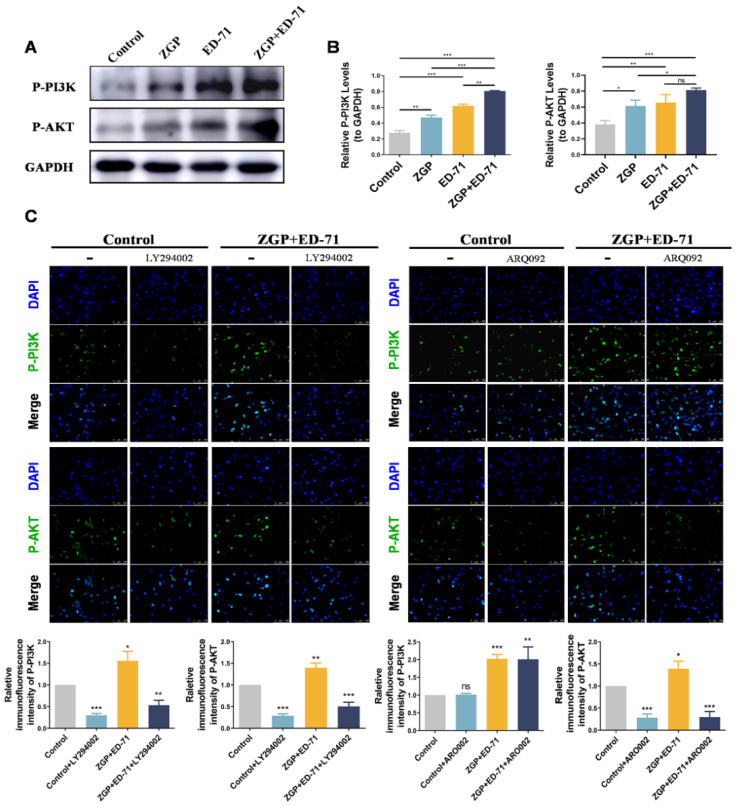
ZGP synergizes with ED-71 in regulating MC3T3-E1 cells through the PI3K–AKT pathway: (**A**) the protein expression of P-PI3K, P-AKT, and GAPDH was detected via Western blotting after 14 days of culture; (**B**) statistical analysis of the Western blot results; (**C**) representative immunofluorescence staining images of P-PI3K and P-AKT in the CON and ZGP + ED-71 groups seven days after adding the small-molecule inhibitors LY294002 and ARQ092, and the quantitative statistical analysis of the immunofluorescence staining. Bar, 200 μm. The data are presented as means ± SDs (*n* = 3). ns: no significance, * *p* < 0.05, ** *p* < 0.01, and *** *p* < 0.001.

**Figure 6 medicina-59-01414-f006:**
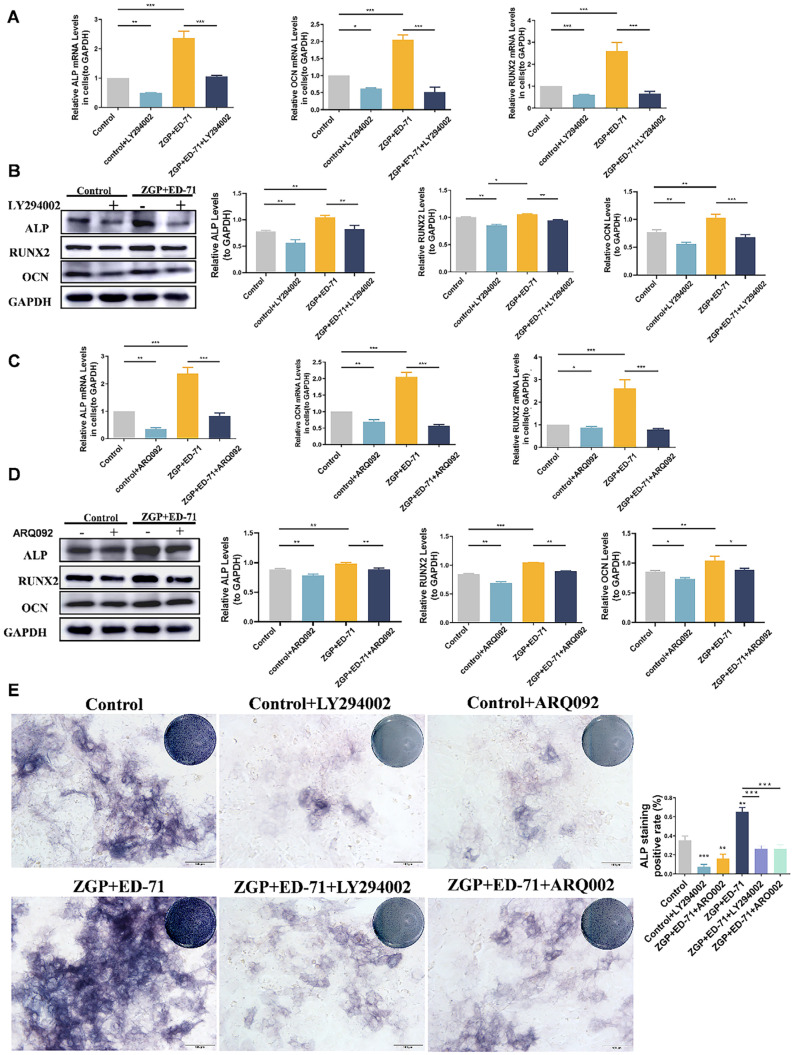
ZGP synergizes with ED-71 in regulating MC3T3-E1 cells through the PI3K–AKT pathway: (**A**,**C**) the *ALP*, *RUNX2*, *OCN*, and *GAPDH* mRNA levels were assessed via RT-qPCR after seven days of culture with and without the small-molecule inhibitors LY294002 and ARQ092; (**B**,**D**) the ALP, RUNX2, OCN, and GAPDH protein expression was detected via Western blotting after 14 days of culture; (**E**) ALP staining of MC3T3-E1 cells in the various groups after seven days of culture, and the quantitative statistical analysis of ALP staining. Bar, 100 μm. The data are presented as means ± SDs (*n* = 3). * *p* < 0.05, ** *p* < 0.01, and *** *p* < 0.001.

**Table 1 medicina-59-01414-t001:** Primer sequences (5′ to 3′) used for mouse genes.

Gene	Forward	Reverse
*OCN*	CAGAACAGACAAGTCCCACACAG	TCAGCAGAGTGAGCAGAAAGAT
*RUNX2*	TACGACCATGAGATTGGCAGTGA	TATAGGATCTGGGTGCAGGCTGA
*ALP*	GCGACCACTTGAGCAAACATC	CGGCTGATTGGCTTCTTCTT
*GAPDH*	GCACCGTCAAGGCTGAGAAC	TGGTGAAGACGCCAGTGGA

## Data Availability

The data presented in this study are available on request from the corresponding author. The data are not publicly available due to restriction of privacy.

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
