# Peer review of "The Synergistic Effect of Zuogui Pill and Eldecalcitol on Improving Bone Mass and Osteogenesis in Type 2 Diabetic Osteoporosis"

_medicina, 2023, doi:10.3390/medicina59081414_

Round 1

Reviewer 1 Report

Manuscript Number: medicina-2474541

Comments to Authors

The authors in the paper “Zuogui Pill and Eldecalcitol synergistically improve type 2 diabetic osteoporosis by promoting osteoblast differentiation through PI3K/AKT signalling pathway” investigated the combined effect of Zuogui Pill and eldecalcitol on improving type diabetic osteoporosis (T2DOP) and explored their action mechanism.

Although interesting, the authors in the manuscript did not explain the composition of the used pills. What is the composition of the pills? What proportion of the mentioned extracts/plants was? Although widely used in CTM, it does not mean that the compatibility with European law exists, especially as the EMA gave specific monographs for the plants used.

Please, explain what does the “nourishing kidney yin” mean (page 2, line 69) – what kind of changes in kidney does that phrase define?

Authors are asked to give the full Latin name of all plants mentioned in the manuscript that represent the composition of the used Zuogui pills in the experiments, the ratio of the plants in the formulation, as well as which part of the plant was used for each plant, and if the extracts were used, the method of preparation of the mentioned extracts, if they were quantified/standardized on active component, or marker compound.

 Please, give the chemical composition of the used Zuogui pills.

Please, give the chemical composition of the used pills, and the adjuvants used to make the pills.

In Materials and method the effect of non-actives, but adjuvants, if existing in the used formulation, should be examined as well

Please, explain the used doses in the experiments, page 3, lines 133,134. Give the scientific explanation why the mentioned doses were chosen. In addition, why was the experiment performed with one dose?

What were the criteria for statement that the joint effect of investigated Zuogui pills and Eldecalcitol might be considered as synergic effect? Although significant improvement, that does not mean that the effect is synergistic.

A lot of work has been done on osteoporosis when Zuogui pills and Eldecalcitol were administrated and all the trial were performed in clinical experiments, giving the data about the efficacy. Why was the study conducted on mice when there are a lot of clinical trials for both components?

Author Response

The responses to the reviewer 1 are as follows.

The authors in the paper “Zuogui Pill and Eldecalcitol synergistically improve type 2 diabetic osteoporosis by promoting osteoblast differentiation through PI3K/AKT signalling pathway” investigated the combined effect of Zuogui Pill and eldecalcitol on improving type diabetic osteoporosis (T2DOP) and explored their action mechanism.

Response: Thanks very much for your affirmation of our work. We are very glad to have good communication with you. We have read your comments one by one, which helps us to improve our study and expand our thinking. Please find our responses to your comments below. The corresponding changes in the manuscript have also been highlighted in red.

Although interesting, the authors in the manuscript did not explain the composition of the used pills. What is the composition of the pills? What proportion of the mentioned extracts/plants was? Although widely used in CTM, it does not mean that the compatibility with European law exists, especially as the EMA gave specific monographs for the plants used.

Response: Many thanks to your comment. The ZGP used in this experiment was a proprietary Chinese medicine purchased from Beijing Tongrentang Co., LTD., rather than extracted by ourselves. It comprises eight Chinese herbs, namely Rehmanniz glutinosa Libosch. (prepared root), Lycium barbarum L. (dried fruit), Dioscorea opposita Thunb. (dried rhizome), Cyathula officinalis Kuan (dried root), Cornus officinalis Sieb. et Zucc. (dried fruit), Cervus elaphus Linnaeus (antler), Cuscuta chinensis Lam. (dried seed), and Chinemys reevesii (Gray) (shell) (8:4:4:3:4:4:4:4) [1]. We tried but failed to obtain more details from the merchant because of commercial confidentiality. In this study, we explored the role and mechanism of ZGP combined with ED-71 in T2DOP treatment from the basic experimental level, and there is still a long way to go before their combined clinical application.

Reference:

[1] Liu FX, Tan F, Fan QL, Tong WW, Teng ZL, Ye SM, Li X, Zhang MY, Chai Y, Mai CY. Zuogui Wan improves trabecular bone microarchitecture in ovariectomy-induced osteoporosis rats by regulating orexin-A and orexin receptor. J Tradit Chin Med. 2021 Dec;41(6):927-934.

Please, explain what does the “nourishing kidney yin” mean (page 2, line 69) – what kind of changes in kidney does that phrase define?

Response: Thank you very much for the valuable opinion. The kidney in TCM is different from the anatomic kidney in modern medicine. According to TCM, the kidney, which is one of the five zang organs, has one of the functions of governing bones. The kidney stores yin essence, the yin essence produces and generates marrow, while the marrow fulfills and nourishes bones. Therefore, when kidney yin essence is deficient and there is no enough nourishment, bone loss, namely osteoporosis, happens. Thereby, nourishing kidney yin is considered as a therapeutic method for osteoporosis.

Authors are asked to give the full Latin name of all plants mentioned in the manuscript that represent the composition of the used Zuogui pills in the experiments, the ratio of the plants in the formulation, as well as which part of the plant was used for each plant, and if the extracts were used, the method of preparation of the mentioned extracts, if they were quantified/standardized on active component, or marker compound.

Response: We greatly appreciate the reviewer for the comment. The full Latin name of all plants that make up ZGP, the ratio of the plants in the formulation, and the part of the plant used for each plant, are as follow, “Rehmanniz glutinosa Libosch. (prepared root), Lycium barbarum L. (dried fruit), Dioscorea opposita Thunb. (dried rhizome), Cyathula officinalis Kuan (dried root), Cornus officinalis Sieb. et Zucc. (dried fruit), Cervus elaphus Linnaeus (antler), Cuscuta chinensis Lam. (dried seed), and Chinemys reevesii (Gray) (shell)”. The ZGP used in this experiment was a proprietary Chinese medicine purchased from Beijing Tongrentang Co., LTD., rather than extracted by ourselves. According to the ancient literature [1], the ratio of the above parts is 8:4:4:3:4:4:4:4.

Please, give the chemical composition of the used Zuogui pills.

Response: Thanks a lot for the comment and we admire your professionalism and deep understanding of the chemical composition of drugs. Due to the effective ingredients of TCM prescriptions are complicated, we reviewed the literature [2] and found that ZGP contains the following active ingredients, such as gallic acid, 5-hydroxymethylfurfural (5-HMF), morroniside, sweroside, loganin, β-ecdysterone, rutin, hyperoside, quercetin, kaempferide etc.

Reviewer 2 Report

The major finding in the current study is the synergistic beneficial effect of Zuogui Pill (ZGP) and eldecalcitol on glycemia levels and bone homeostasis in vivo, and on osteoblast differentiation in vitro. This finding is based on the previous studies from other groups showing the similar effects produced by ZGP and eldecalcitol alone.  The research is properly designed and is somewhat comprehensive.  The major deficiency is that ZGP was currently studied as a mixed extract of herbal materials without identifying the active single ingredient though a few active compounds are mentioned in the “Discussion”.  In addition, the writing needs to be improved in term of scientific rigor, research method description, and sentence fluency and wording in English. 

1.    In a number of places in the manuscript, the authors claim that the treatment altered the osteoclast and osteoblast activity in vivo based on the staining of the cell markers.  This claim is too stretching without directly measuring the activity of the bone cells.  Similarly, it is claimed that the treatment promoted new bone regeneration solely based on Masson staining (Lines 238-239).  Actually, the increase in Masson staining may not be just due to the increase in regeneration, but could also be the results of the reduction in degradation.

2.    Lines 315-317:  “…..mainly achieved by promoting osteoblast differentiation.”  This statement is biased.  The outcome of ZGP and eldecalcitol on bone could be resulted from multiple mechanisms of action, including the action on osteoclast.  In this study, only osteoblastic effect has been mainly investigated and the other mechanisms could not be excluded.

3.    It should be indicated what vehicle was used in the control group.  Eldicalcitol was dissolved in 100% ethanol (Line 117).  What was used to further dilute it and was ethanol included in the control?

4.    What secondary antibody was used?  Please specify it.

5.    Line 176: “Conditioned medium” usually refers to the medium obtained from previous cell cultured.  Do you really mean this?  The blood glucose levels in the mice was about 20 mM.  Why much higher glucose concentration (35 mM) was used in the cell culture?

6.    How calcium and phosphor were measured (wrongly spelt as “calucium” and “phosphuros” in Table 2D)?  This should be described.

7.    The captions in many figures are too small.

8.    Figures 2A and 5C: What treatment for each row of the images?

9.    Figure 3B: How were the values in the figures obtained?  Please describe this.

10.  Line 103: Ref 30 was wrongly cited.  Since the ref order was wrong, the subsequent refs could also be wrongly cited.  Please double check carefully.

11.  Line 158: Reading digital pictures by using optical microscope?  It doesn’t make sense.

12.  Lines 370-372, 379-380: These sentences do not make sense.

Please see the comments above.

Author Response

The responses to the reviewer 2 are as follows.

The major finding in the current study is the synergistic beneficial effect of Zuogui Pill (ZGP) and eldecalcitol on glycemia levels and bone homeostasis in vivo, and on osteoblast differentiation in vitro. This finding is based on the previous studies from other groups showing the similar effects produced by ZGP and eldecalcitol alone. The research is properly designed and is somewhat comprehensive. The major deficiency is that ZGP was currently studied as a mixed extract of herbal materials without identifying the active single ingredient though a few active compounds are mentioned in the “Discussion”. In addition, the writing needs to be improved in term of scientific rigor, research method description, and sentence fluency and wording in English.

ResponseThank you for your kind comments concerning our manuscript. We have carefully reviewed the comments and exerted all our energies to revise the manuscript according to the suggestions, which are highlighted red in the manuscript. We sincerely hope that our revision will meet with your approval.

  1. In a number of places in the manuscript, the authors claim that the treatment altered the osteoclast and osteoblast activity in vivo based on the staining of the cell markers. This claim is too stretching without directly measuring the activity of the bone cells. Similarly, it is claimed that the treatment promoted new bone regeneration solely based on Masson staining (Lines 238-239). Actually, the increase in Masson staining may not be just due to the increase in regeneration, but could also be the results of the reduction in degradation.

ResponseThank you for your constructive criticism. Considering our in vivo IHC staining and quantitative analysis results showed changes in the levels of ALP and RUNX2 (markers of osteoblast differentiation), COL-1 (a marker of osteoblast function) and MMP9 (an enzyme secreted by osteoclasts), we considered that the treatment altered osteoblast and osteoclast activity in vivo. Based on the comment reviewer have given, we have rephrased it and removed the word "activity" to avoid misunderstanding. As for the Masson staining, previous studies have suggested that the blue-stained bone tissue can be used to show new bone formation [1]. Therefore, combining the results of HE staining and Masson staining, we concluded that the treatment increased bone mass and improved new bone formation. We have also rewritten the description in the Results, making it easier to understand.

Reference:

[1] Chen MH, Hu Y, Hou YH, Sun YT, Chen MW, Li MH, et al. Construction of a reactive oxygen species-responsive biomimetic multilayered titanium implant for in situ delivery of alpha-melanocyte-stimulating hormone to improve bone remolding in osteoporotic rats. Applied Materials Today. 2021;23.

  1. Lines 315-317: “…..mainly achieved by promoting osteoblast differentiation.” This statement is biased. The outcome of ZGP and eldecalcitol on bone could be resulted from multiple mechanisms of action, including the action on osteoclast. In this study, only osteoblastic effect has been mainly investigated and the other mechanisms could not be excluded.

ResponseThank you for your professional comment, which has greatly benefited us. We realized that our previous statement was too absolute, so we rewrote here, changing “mainly” to “at least in part”.

  1. It should be indicated what vehicle was used in the control group. Eldicalcitol was dissolved in 100% ethanol (Line 117). What was used to further dilute it and was ethanol included in the control?

ResponseThanks for your valuable comments. We have clearly supplemented that ED-71 was dissolved in 100% ethanol for storage and was diluted to the right concentration by medium chain triglycerides (MCTs) before usage, and MCTs containing the same concentration of absolute ethanol were used as controls. We have supplemented these contents in detail in the Materials and Methods.

  1. What secondary antibody was used? Please specify it.

ResponseThanks for your helpful suggestion. We have enumerated the secondary antibodies used, including “goat anti-rabbit (ab6721), goat anti-mouse (SA00001-1), and goat anti-rabbit (SA00013-2) IgGs”, in the Materials and Methods.

  1. Line 176: “Conditioned medium” usually refers to the medium obtained from previous cell cultured. Do you really mean this? The blood glucose levels in the mice was about 20 mM. Why much higher glucose concentration (35 mM) was used in the cell culture?

ResponseWe greatly appreciate you for the comment. In our study, the “Conditioned medium” refers to the osteogenesis induction medium which contained the serum from different groups of administered rats. We have made references to your valuable comments, and in order to avoid misunderstanding, we have deleted the “Conditioned” and changed to “osteogenesis-inducing medium”. It has been reported that the glucose concentration required to simulate a hyperglycemic environment in vitro is 35 mM [2], and we used this optimal concentration in order to successfully simulate a hyperglycemic environment in vitro. We have supplemented the reference in the manuscript, thanks again.

Reference:

[2] Liu B, Gan X, Zhao Y, Gao J, Yu H. Inhibition of HMGB1 reduced high glucose-induced BMSCs apoptosis via activation of AMPK and regulation of mitochondrial functions. J Physiol Biochem. 2021;77(2):227-35.

  1. How calcium and phosphor were measured (wrongly spelt as “calucium” and “phosphuros” in Table 2D)? This should be described.

ResponseThanks for your careful review of the article, we have corrected this spelling error. As for the specific measurement of calcium and phosphor, we have supplemented these contents in detail in the Materials and Methods.

  1. The captions in many figures are too small.

ResponseThank you for your valuable comments. We have enlarged the size of the captions in the figures to make it easier to read the pictures and our manuscript.

  1. Figures 2A and 5C: What treatment for each row of the images?

ResponseThanks for bringing up these issues. In Figure 2A and 5C, we have annotated the specific treatment shown by each row of pictures.

  1. Figure 3B: How were the values in the figures obtained?  Please describe this.

ResponseThanks a lot for the valuable opinion. The immunohistochemical positive expression was be quantitatively evaluated by optical density. In our study, all quantitative analyses of immunohistochemical staining were performed using IPP software, and they were obtained by using positive expression intensity in the region/area of the region. Especially, we properly used the number of RUNX2-positive cells in the quantitative analyses of RUNX2, because the expression of RUNX2 was concentrated in the nucleus of positive cells [3].

Reference:

[3] Loncoñanco, Ester, Navarrete, Felipe, Cuevas, Nicolás, Vasconcellos, Adriana, & Paredes, Marco. Integrated Optical Density Analysis of the Immunohistochemical Expression of the Progesterone Receptor in the Uterine Endometrium of Prepubertal Araucana Sheep. International Journal of Morphology. 2021;39(5):1278-1282.

  1. Line 103: Ref 30 was wrongly cited. Since the ref order was wrong, the subsequent refs could also be wrongly cited. Please double check carefully.

ResponseWe are sorry for our mistake. We have double checked the reference carefully, and reference 30 was changed to the correct format. Besides, we have rechecked the subsequent references to make sure they were referred correctly.

  1. Line 158: Reading digital pictures by using optical microscope? It doesn’t make sense.

ResponseThank you for your kind comments. The optical microscope we used is capable of observing the sections and taking pictures. To avoid ambiguity, we have modified the statements in the Materials and Methods.

  1. Lines 370-372, 379-380: These sentences do not make sense.

ResponseWe really appreciate your careful comments, and we apologize for these two ambiguous sentences. We have rephrased these sentences in the manuscript. We hope the modifications will meet with your full satisfaction.

Reviewer 3 Report

The paper entitled “Zuogui Pill and Eldecalcitol synergistically improve type 2 diabetic osteoporosis by promoting osteoblast differentiation through PI3K/AKT signaling pathway” is presented. The authors tested the therapeutic effect of Zuogui Pill and Eldecalcitol in mice with type 2 diabetic osteoporosis. They showed the combined effect of Zuogui Pill and Eldecalcitol on type 2 diabetic osteoporosis through PI3K/AKT mediated signaling pathway in osteoblasts. The study is interesting and well-organized.

There are some possible issues

Figure 4C-D. Can you provide quantitative data for this?

Figure 5 C. Can you provide quantitative data for this?

Figure 6E. Can you provide quantitative data for this?

Fig3. Do Zuogui Pill and Eldecalcitol affect osteoclasts? In this paper, Zuogui Pill and Eldecalcitol might have multiple effects like TCM.  Many studies have found Therapeutic Potential and Outlook of Alternative Medicine in bone conditions via pro-anabolic effects and anti-catabolic effects and the regulation of osteoclasts and osteoblasts (for example PMID: 28325144, PMID: 31824310). It would be relevant to discuss the pro-anabolic osteoblastic effects and anti-catabolic osteoclastic effects of Zuogui Pill and Eldecalcitol on bone.

It was mentioned that Zuogui Pill and Eldecalcitol affect PI3K/AKT signaling pathways, which are also important in osteoclast formation (for example, PMID: 35988868, PMID: 30947099, PMID: 26638989). It would be informative to discuss the possible role of Zuogui Pill and Eldecalcitol in PI3K/AKT signaling pathways in osteoclasts as suggested.  

Reviewer 4 Report

The manuscript is well-written; however, there are a few aspects that could be addressed for further improvement:

1.       The authors have convincingly demonstrated the synergistic promotion of osteoblast activity and inhibition of osteoclast activity in a diabetic animal model through the administration of ZGP and Ed-71. While the authors have provided valuable insights by performing immunohistochemical (IHC) analysis of MMP9 to depict the change in osteoclast activity, it would be beneficial to supplement this analysis with TRAP staining of bone tissue sections. This additional experiment would further validate the significant inhibition of osteoclast activity resulting from ZGP+Ed-71 treatment.

2.       In Figure 2, both in vivo and in vitro experiments were conducted using diabetic mice or osteoblast cells cultured in high glucose media, and subsequently treated with ZGP and Ed-71 to evaluate their effects on bone remodeling and osteoblast (OB) activity. To enhance the comprehensiveness of the study, it would be advantageous to include control groups consisting of wild-type (non-diabetic) animals or normal cells (OB cells cultured in non-high glucose media) for comparison. This inclusion serves two purposes: firstly, it would provide insights into how ZGP and/or Ed-71 impact the non-diabetic control groups, and secondly, it would enable an assessment of whether the effects of these drugs on diabetic animals/cells represent complete or partial recovery in comparison to the wild-type or normal controls. In essence, it would elucidate whether the treatment of ZGP+Ed-71 restores osteoblast activity and bone mass to levels comparable to those observed in the wild-type controls.

3.       Figure 1 effectively demonstrates the ability of ZGP and Ed-71 to reduce glucose levels in diabetic animals. However, it is worth noting that the manuscript lacks experimental investigations or discussions pertaining to the relationship between changes in glucose levels induced by these drugs and alterations in bone cell differentiation, activity, and related factors. To provide a more comprehensive understanding, it would be valuable to explore or discuss the potential associations between the observed changes in glucose levels and the effects on bone cell differentiation and activity resulting from ZGP and Ed-71 treatment.

Author Response

The responses to the reviewer 4 are as follows.

The manuscript is well-written; however, there are a few aspects that could be addressed for further improvement:

Response: Thanks very much for your affirmation of our work. We have read the comments one by one, which helps us to improve our study and expand our thinking. Please find our responses to your comments below. The corresponding changes in the manuscript have also been highlighted in red.

  1. The authors have convincingly demonstrated the synergistic promotion of osteoblast activity and inhibition of osteoclast activity in a diabetic animal model through the administration of ZGP and Ed-71. While the authors have provided valuable insights by performing immunohistochemical (IHC) analysis of MMP9 to depict the change in osteoclast activity, it would be beneficial to supplement this analysis with TRAP staining of bone tissue sections. This additional experiment would further validate the significant inhibition of osteoclast activity resulting from ZGP+Ed-71 treatment.

Response: Thanks very much for your comment. We have added TRAP staining of bone tissue sections, and the results showed that ZGP and ED-71 could inhibit the number of TRAP-positive osteoclasts, especially when combined. We have also added the corresponding content in the figure and manuscript.

  1. In Figure 2, both in vivo and in vitro experiments were conducted using diabetic mice or osteoblast cells cultured in high glucose media, and subsequently treated with ZGP and Ed-71 to evaluate their effects on bone remodeling and osteoblast (OB) activity. To enhance the comprehensiveness of the study, it would be advantageous to include control groups consisting of wild-type (non-diabetic) animals or normal cells (OB cells cultured in non-high glucose media) for comparison. This inclusion serves two purposes: firstly, it would provide insights into how ZGP and/or Ed-71 impact the non-diabetic control groups, and secondly, it would enable an assessment of whether the effects of these drugs on diabetic animals/cells represent complete or partial recovery in comparison to the wild-type or normal controls. In essence, it would elucidate whether the treatment of ZGP+Ed-71 restores osteoblast activity and bone mass to levels comparable to those observed in the wild-type controls.

Response: Many thanks to the comment and we admire your professionalism and deep understanding. In our previous study, we found that WT mice and diabetic mice have a large difference in bone mass, and diabetic mice have significant bone mass loss [1]. In this study, we focused on the effect of ZGP+ED-71 combination treatment on diabetic osteoporosis, which is why we selected the diabetic mice and high glucose environment. With your valuable comments in mind, we are committed to further clarifying the effects of the combination on non-diabetic animals and cells in future studies and to assess the extent of bone mass recovery after the combination. If you have any more insights, please contact us for further communication, thanks again.

Reference:

[1] Lu Y, Liu S, Yang P, et al. Exendin-4 and eldecalcitol synergistically promote osteogenic differentiation of bone marrow mesenchymal stem cells through M2 macrophages polarization via PI3K/AKT pathway. Stem Cell Res Ther. 2022;13(1):113. Published 2022 Mar 21. doi:10.1186/s13287-022-02800-8

  1. Figure 1 effectively demonstrates the ability of ZGP and Ed-71 to reduce glucose levels in diabetic animals. However, it is worth noting that the manuscript lacks experimental investigations or discussions pertaining to the relationship between changes in glucose levels induced by these drugs and alterations in bone cell differentiation, activity, and related factors. To provide a more comprehensive understanding, it would be valuable to explore or discuss the potential associations between the observed changes in glucose levels and the effects on bone cell differentiation and activity resulting from ZGP and Ed-71 treatment.

Response: Thank you for your constructive suggestions, which really inspired us a lot. We highly agree with your opinion that there are potential associations between the reduced-glucose levels and the effects on bone cell differentiation and activity resulting from ZGP and ED-71 co-treatment. This can also be confirmed in the previous study which shows that high glucose environment can inhibit the osteogenic differentiation and activity of osteoblasts as well as increase the osteoclast differentiation and osteoclast activity [2]. We really appreciate your comprehensive comments, and we have supplemented these contents in the Discussion.

Reference:

[2] Rathinavelu S, Guidry-Elizondo C, Banu J. Molecular Modulation of Osteoblasts and Osteoclasts in Type 2 Diabetes. Journal of Diabetes Research. 2018;2018.

Round 2

Reviewer 1 Report

Article: medicina-2474541  

Title: Zuogui Pill and Eldecalcitol synergistically improve type 2 diabetic osteoporosis by promoting osteoblast differentiation through PI3K/AKT signaling pathway

Although authors improved and answered some questions, still remained to incorporate the explanation given to reviewer into the main text. As this manuscript is not attended only for Chinese medicine consumers and experts, the terms from TCM should be explained in the understandable terms. That refers to “nourishing kidney yin”, and the explanation should be given in the main text.

As authors were asked to give the full Latin name of all plants/animals mentioned in the manuscript that represented the composition of the used Zuogui pills in the experiments, please, give Families of each components present in the form. In addition, what does it mean: Rehmanniz glutinosa Libosch. - prepared root? How was it prepared?

In addition, the plants and the turtle shell were mixed and put together into honey - what is the percentage of the actives in the honey?

Although known under the trade name, Zuogui pills used in the experiments should be chemically characterised, and the investigation should not be based on the specification given in some previous work. 

Usually, when the confirmation of some traditionally used preparation is the aim of the work, the correlation between the chemical composition and the observed activity is necessary. As traditionally used, it is known to be effective, and the explanation "why" is lacking. So, nevertheless that you purchased the medicine in shop, the chemical characterisation have to be done.

Reviewer 2 Report

1.  Lines 97-98: Ref 30 is still incorrectly cited.  This is not your previous works, but the works from the other group.

2.  The caption in some figures in Figure 6 is still too small.

Reviewer 4 Report

Thank you for addressing the comments.  I will be fine for the manuscript to be accepted in present form.

Author Response

ResponseThank you very much for your review of our manuscript and thanks for your recognition of our work. Best wishes to you.